# Beliefs and Self-Perceptions of Spanish Mental Health Professionals about Physical Therapy in Mental Health: An Observational Survey Study

**DOI:** 10.3390/healthcare11243136

**Published:** 2023-12-11

**Authors:** Cristina Bravo, Emilio Minano-Garrido, Lidia Carballo-Costa, Miguel Muñoz-Cruzado y Barba, Silvia Solé, Francesc Rubí-Carnacea, Daniel Catalan-Matamoros

**Affiliations:** 1Group of Salut&Genesis, Campus de Salud UdL-Igualada, 08700 Barcelona, Spain; cristina.bravo@udl.cat; 2Department of Nursing and Physiotherapy, University of Lleida, 25006 Lleida, Spain; francesc.rubi@udl.cat; 3Grup d’Estudis Societat, Salut, Educació i Cultura, GESEC, Department of Nursing and Physiotherapy, University of Lleida, 25006 Lleida, Spain; 4Health Care Research Group (GRECS), Lleida Institute for Biomedical Research Dr. Pifarré Foundation, IRBLleida, 25198 Lleida, Spain; 5International School of Doctoral Studies, University of Murcia, 30100 Murcia, Spain; emiliojose.minanog@um.es; 6Department of Physical Therapy, Hôpital Haut-Lévêque, CHU Bordeaux, F-33000 Bordeaux, France; 7Grupo de investigación en Intervención psicosocial y Rehabilitación funcional, Faculty of Physiotherapy Universidade da Coruña, Campus de A Coruña, 15071 A Coruña, Spain; lidia.carballo@udc.es; 8Department of Physiotherapy, Faculty of Health Sciences, University of Malaga, 29016 Malaga, Spain; mmunozcb@gmail.com; 9Research Group CTS 451 “Health Sciences”, University of Almeria, 04120 Almeria, Spain; dcatalan@ual.es; 10Science/Health Communication, University Carlos III of Madrid, 28911 Madrid, Spain

**Keywords:** physiotherapy, mental health, therapeutic exercise, body awareness, therapeutic alliance

## Abstract

Objective: The aim of this study is to understand the image, perception, and beliefs regarding the role of the physiotherapist in the field of mental health physiotherapy, both among the professional community and other multidisciplinary teams. Methods: An observational phenomenological qualitative study through the administration of an ad hoc survey comprising both categorical and open-ended as well as quantitative questions was conducted. Results: A total of 368 responses were analysed. The participants comprised 78.4% women with a mean age of 37.5, an average professional experience of 14.33 years, and 88.3% practicing physical therapists. From the qualitative analysis conducted, three categories emerged in relation to the obtained responses: (a) functions with codes of “improving quality of life” and “intervening in physical pathologies”; (b) objectives with codes of “Improving quality of life”, “Intervening in physical pathologies”, “Functional rehabilitation”, “Health promotion”, and “Intervening in mental disorders”; and (c) image with codes “unfamiliarity”, “holistic vision”, “necessity”, and “importance”. Regarding the tools, the findings highlight a strong focus on physical exercise interventions due to their well-established benefits. Cognitive strategies like therapeutic relationships and cognitive–behavioural techniques were also prominent. Additionally, embodiment techniques involving movement, relaxation, breathing, and voice usage were notable. Lastly, manual therapy and physical agents formed another distinct category. Conclusions: The vision and role of this professional profile were unknown to the respondents. Despite being perceived as having a holistic view of the patient and being considered an essential need, the actual image remains vague. However, there is significant interest, indicating a promising future, although the lack of specialized training is noted. Therefore, the need for specialized education and awareness campaigns among professionals in the mental health field is highlighted.

## 1. Introduction

Physiotherapy in mental health (PTMH) is a specialty of physiotherapy, which is defined as the art and science of applying physical agents for the promotion of individuals’ overall health, including the prevention and treatment of mild, moderate, or severe mental health conditions, as well as acute or chronic conditions, and the rehabilitation of their consequences [1]. The aim is to optimize the physical and mental aspects and enhance the well-being of individuals at any age, empowering them through the promotion of functional movements, body awareness and movement, physical activity, and therapeutic exercise. The physiotherapist in mental health addresses the complexity of disorders by creating an appropriate therapeutic environment and relationship to apply the biopsychosocial model, wherein the focus is on the person, including human rights, within the context of rehabilitation. Drawing upon the latest international scientific and clinical evidence, the physiotherapist in mental health can practice in primary and community care, in inpatient and outpatient settings, in public and private centres, and in social institutions, contributing benefits to individuals cared for by the transdisciplinary mental health team, wherein the physiotherapist plays a prominent role [1].

Physical and mental health conditions coexist, and they should not be treated in isolation [2]. Psychiatry and mental health perceptions have been influenced by philosophy and psychoanalysis, yielding concepts like somatization, which is related to poor physical outcomes. Psychocorporal and somatic approaches were developed by Norwegian physiotherapists like Trygve Braatoy and Bülow-Hansen, pioneers in Nordic body awareness approaches such as “Norwegian Psychomotor Physiotherapy” and “Basic Body Awareness Therapy”, which are currently used in mental health physiotherapy [3].

According to studies conducted in Australia [4,5], physiotherapists feel prepared to manage the physical health of patients with mental disorders, although they request more undergraduate education on mental health and that the health system barriers are addressed. Another study [6] conducted with the International Organization Physical Therapists in Mental Health (IOPTMH) members on the role of physiotherapy in the treatment of patients suffering from schizophrenia concluded that physiotherapists are an integral part of the multidisciplinary team that is focused on promoting the physical health needs of these patients.

Physiotherapy emerged as a nursing specialty because of another polio epidemic in the 1950s. It was not until 1999 that the World Confederation of Physical Therapist (WCPT) defined it as an autonomous profession capable of performing assessment, diagnosis, planning, intervention, and evaluation, constituting the essence of its professional identity. In 1982, the Physiotherapy Degree was established with the publication of the Royal Decree on 26 July, and in 2007, it was implemented as an undergraduate degree, providing access to master’s and doctoral programmes. Currently, physiotherapy is offered as Higher Education studies in most countries worldwide [7,8].

Specialties in physiotherapy have been proposed for recognition but without success so far. Specialization in physiotherapy is defined as a professional with advanced clinical competencies in a specific and defined area of the profession. This professional requires academic training with experience, clinical supervision, and development of professional competencies to apply, mediate, and develop specialized knowledge [8]. Currently, specialties in physiotherapy exist in Austria, Denmark, Finland, Germany, Italy, the Netherlands, and the United Kingdom. In this country, local areas of the National Health System (NHS) offer PTMH in their service directory [9]. Both the NHS and the Chartered Society of Physiotherapy (CSP) have published documents describing the requirements to become a specialist, as well as policy documents and treatment guides regarding PTMH [10,11]. In Spain, physiotherapists in mental health settings primarily address musculoskeletal or neurological conditions in patients. However, the specific goal of alleviating mental health symptoms within this specialty is not integrated into the mental health team. Regarding training, there are postgraduate courses and master’s programmes that incorporate subjects related to mental health physiotherapy to ensure the high-quality training of physiotherapists who would work in this field.

PTMH has been evolving in recent years and includes a wide variety of evidence-based techniques [12], such as physical activity and body awareness therapies, that have been shown to improve both physical and mental symptomatology, as well as the quality of life for individuals with various types of mental disorders such as depression, anxiety, eating disorders, and schizophrenia [2,13,14]. In countries like Sweden, Norway, England, and Belgium, mental health physiotherapy has been developing both theoretically and practically and has gained importance in the mental health services of these countries [15], as well as in their university training programmes [12]. Physical therapists in general practice settings recognize the need for more training to enhance their work with people with mental health comorbidities [2]. However, proactive efforts are still needed in local and national policies and influencing decision-making committees to raise awareness of the benefits that this specialty can offer.

According to the World Health Organization (WHO), before the COVID-19 pandemic, more than one billion people were already experiencing some form of mental disorder, with 82% of these individuals residing in low- or middle-income countries, making access to treatment even more challenging. Following the pandemic, these figures have significantly increased, with the WHO estimating that disorders such as anxiety and depression have risen by 25% to 27% [16].

Individuals with severe mental disorders (e.g., schizophrenia, bipolar disorder, and major depressive disorder) also face a higher risk of experiencing a myriad of comorbidities [17], resulting in a life expectancy that is 15 to 25 years lower than the general population [18,19]. Consequently, mental health disorders pose a significant threat to public health and have become a critical issue in both national and international policies [20].

Hence, there arises a need to understand the image, perception, and beliefs regarding the role of the physiotherapist in the field of PTMH in Spain, both from the professional community and the rest of the multidisciplinary team. This understanding will help to identify preconceived notions about this area and establish the foundations for its development in terms of employment, education, and research. Therefore, the objective of our study is to analyse, understand, and comprehend the perceptions and beliefs held by professionals in the mental health field, and physiotherapists themselves, concerning the concept of mental health physiotherapy.

## 2. Methodology

### 2.1. Study Selection

An observational phenomenological qualitative study through the administration of an ad hoc survey comprising both categorical and open-ended as well as quantitative questions was conducted in June 2022.

### 2.2. Instrument and Data Extraction

To achieve the study’s objective, various research questions (Table 1) were established, encompassing sociodemographic information, professional activity, mental health training, and perceptions, beliefs, image, and role of the physiotherapist in mental health. The survey was designed through the consensus of various mental health experts who analysed the different questions over a period of one month to address the research question. All relevant variables were identified, and open-ended questions were crafted to gather the maximum possible data.

### 2.3. Participants

The survey was sent to a convenience sample of professionals working in the field of mental health. The participants were selected according to the inclusion criteria: (a) a group of physiotherapists from Spain, (b) different professionals involved in the treatment of mental disorders as part of a multidisciplinary team (psychiatrists, psychologists, nurses, social workers, occupational therapists). Those who did not work or live in Spain and those who did not give their consent to participate were excluded from the survey.

Emails were sent to various national associations and professional bodies of professionals related to mental health in early May. Additionally, dissemination was carried out through official social media channels of the Spanish Association of Physiotherapists in Mental Health (AEF-SM) and members of the board. Furthermore, all members of the research team were requested to collaborate by spreading the survey link through WhatsApp, LinkedIn, and Twitter groups, primarily. A survey link was provided through Google Forms, which was completed by the participants.

### 2.4. Data Synthesis and Analysis

The data were collected through a Google Forms questionnaire and subsequently analysed. The data analysis was descriptive, involving the assessment of relative frequencies in percentage (%), and Microsoft Excel was used for this purpose. For qualitative analysis, Atlas.ti software was employed.

The research rigor was ensured by adhering to the COREQ criteria [21] for the presentation of qualitative research. Throughout the process, Guba and Lincoln’s rigor criteria [22] were implemented, triangulating data generation techniques (discourse technique and representation technique), conducting a literature search, and involving triangulation among three independent researchers. These three researchers, who independently analysed the data, are part of the research team for this study, and they are also doctoral candidates and physiotherapists with prior experience in qualitative studies and proficiency in using Atlas.ti (https://atlasti.com/). The verbatim transcripts of the writings were submitted for validation by the participants.

The data were analysed until saturation was reached by the three researchers, providing an interdisciplinary perspective, and until reaching consensus on the results. Each of the involved researchers independently conducted open coding of the interviews, enabling triangulation of the data analysis. This process began with open coding in relation to the research questions, while also allowing for the exploration of emerging themes. From this coding, categories were generated, which were then grouped into more inclusive ones. Subsequently, an analysis was conducted to identify relationships between categories and patterns, followed by an interpretative phase. Finally, the results’ report was prepared (Figure 1).

## 3. Results

The survey “The Role of Physiotherapy in Mental Health in Spain” was launched by AEF-SM in May 2022, and data from participants were collected from 5 May to 14 June, 2022, when the survey was closed. A total of 368 responses were analysed. Participants received the information through various associations, as depicted in Figure 2. One participant from outside Spain and one who did not consent to participating were excluded. The participants were 78.4% women with a mean age of 37.5, an average professional experience of 14.33 years, and 88.3% being practicing physical therapists. The sociodemographic data of the participants are presented in Table 2.

Regarding the participants’ work areas, they worked in several areas simultaneously, with the most common ones being musculoskeletal physiotherapy, neurology, geriatrics, paediatrics, respiratory, and mental health, and mental health (Table 3).

Finally, the survey respondents received the survey information through the following channels: physiotherapy professional associations (66%), WhatsApp (9.5%), and through other colleagues (6.25%), primarily.

### 3.1. Qualitative Data

The coding of responses was performed globally, generating the same groups of codes. From the analysis conducted, three categories emerged in relation to the obtained responses: (a) functions, (b) objectives, and (c) image. The categories “Functions” and “Objectives” shared codes, as some responses referred to functions already performed by the physiotherapist, while others proposed them as work objectives. Some of the most prominent responses are displayed in Table 4.

### 3.2. Objectives of Physical Therapy in Mental Health

The codes derived within the “Objectives” category, in order of frequency, were as follows: Improving quality of life, Intervening in physical pathologies, Functional rehabilitation, Health promotion, Intervening in mental disorders, Pain relief, Promoting autonomy, Addressing musculoskeletal conditions, and Alleviating symptoms, with 32 citations. These findings reveal three main blocks concerning the objectives of PTMH: enhancing quality of life, intervening in mental disorders, and addressing physical pathologies. Considering the frequency of appearance of these categories, the objectives seem to focus more on the intervention of physical pathologies rather than purely addressing mental disorders. The predominant concept revolves around improving the quality of life through functional rehabilitation and health promotion. The direct intervention to alleviate symptoms related to mental health does not show significant prevalence (Figure 3).

### 3.3. Image of Physical Therapy in Mental Health

In the category “Image”, 17 codes emerged, with the main one being “unfamiliarity”, followed by “holistic vision”, “necessity”, and “importance”. This indicates that within our group of physiotherapists, this role is not well-known, but there is an understanding that its primary characteristic is a comprehensive approach to patient care, recognizing the necessity and importance of this vision that allows for addressing all aspects influencing people’s health.

The unfamiliarity with this field of work also influences the perception of being under-recognized, of limited utility, marginalized, and underdeveloped. The respondents were unfamiliar with the vision and role of physiotherapists in mental health. PT consider the perception of their role in mental health as essential as well as the holistic perspective of the patient.

Regarding the code “training”, there were different opinions about whether sufficient training existed or not. Two citations expressed a lack of necessary training to work in this field. This fact can be related to the code “underdeveloped”, as the lack of specialized accessible training for physiotherapists in undergraduate and postgraduate programmes may hinder the field’s development. There is no sufficient specialized training in physiotherapy in mental health, despite the substantial interest shown by professionals working in this field.

However, it is evident from the code “interest” that this emerging field in our country generates a certain level of interest, with respondents expressing a “promising future” due to the pending work and opportunities that lie ahead (Figure 4).

As can be seen in Figure 4, we establish relationships between the categories “unfamiliarity” and “limited utility, and marginated” since the latter factor is related to the limited development and the need for training required by this new specialization. On the other hand, many participants expressed their interest and see this specialization as having significant potential in the future.

### 3.4. Functions of Physical Therapy in Mental Health

When asked about the purpose of physiotherapy in the field of mental health, the objective was to identify the functions of the physiotherapist. In general, the most unanimous code for this question was “improving quality of life”. The second code that appeared was “intervening in physical pathologies”, indicating that respondents associated this professional profile more with the physical domain rather than the treatment of mental illnesses. The corresponding codes were “health promotion”, “psychological support”, and “intervening in mental disorders”. This suggests that there is still a lack of awareness about the main scope of PTMH.

The importance of the multidisciplinary team was also emphasized, as respondents understood that they do not work in isolation with the patient. The corresponding codes were “promoting autonomy”, “physical well-being”, “awareness”, “emotional management”, “symptom relief”, “mental well-being”, “mitigating medication effects”, and “somatization” (Figure 5).

### 3.5. Tools of Physical Therapy in Mental Health

In the category “Tools”, the main code that emerged was “physical exercise”, followed by “therapeutic relationship”, “pedagogical approach”, “manual therapy”, “relaxation”, “breathing”, and “movement and movement intervention”. Other codes included “mental health coping tools”, “contact”, and “body awareness”. Additionally, there were mentions of other tools such as electrotherapy, group and individual treatments, neurology, cognitive-behavioural approaches, hydrotherapy, RPG, osteopathy/myofascial techniques, and acupuncture.

The results show a high prevalence of interventions related to physical exercise, aligning with the scientifically validated benefits of physical activity. Other notable tools include those related to cognitive approaches, such as the therapeutic relationship, pedagogical approach, and cognitive-behavioural techniques. Lastly, there is a notable group of tools related to embodiment, such as movement, relaxation, breathing, and voice utilization. In a final grouping, we have tools related to manual therapy or physical agents (Figure 6).

## 4. Discussion

Our study aimed to analyse, understand, and comprehend the perceptions and beliefs held by professionals in the field of mental health and by physiotherapists themselves regarding the concept of PTMH. To achieve this, we conducted a qualitative study, which identified different categories related to objectives, image, functions, and tools in mental health physiotherapy.

By utilizing the data extracted from the survey, we can contextualize the role of the physiotherapist in mental health in Spain. The described image portrays a professional with a significant and necessary holistic approach, yet one that is relatively unknown, marginalized, and underdeveloped due to a lack of adequate training. Regarding therapeutic tools, therapeutic exercise, the therapeutic relationship, and a pedagogical approach with the patient were considered useful.

The importance of a “holistic approach” to patient care is being emphasized in all national and international policies due to the characteristics of patients with mental disorders. Organizations like the WHO are recommending new holistic approaches for the treatment of various mental health conditions at different ages [23]. Therefore, professionals involved in mental health care must be open to other types of interventions, such as physiotherapy.

In mental health physiotherapy, various therapies known as holistic, or body awareness therapies, incorporate psychosocial factors in assessments and treatments, considering them as determining factors for intervention.

Regarding the “lack of awareness” of the specialty by the rest of the multidisciplinary team, it is mainly due to the lack of awareness within our own profession in our country. While other specialties, such as musculoskeletal physiotherapy, have a significant impact and offer numerous training courses and postgraduate programmes, mental health physiotherapy did not have specific training in our country until 2018. The postgraduate training required for specialization is determined by the needs and interests of specialized training centres. This creates a cycle where the lack of awareness about the specialty leads to a lack of interest among physiotherapists, which in turn results in a limited development of specialized courses. It should be noted that there is also no regulated training (official master’s or university-specific programme) for developing the competencies of a mental health physiotherapist in Spain.

The results of the image regarding knowledge gaps align with findings from other studies wherein medical professionals exhibited a lack of knowledge within our field [24]. Even final-year medical students in Nigeria displayed a moderate level of unfamiliarity along with a positive attitude towards mental health [25]. Hence, numerous participants have articulated the notion that the specialty primarily involves the management of physical pathologies rather than mental health concerns.

The categories of “important” and “necessary” stand out in the survey, and this may be attributed to the increasing prevalence of mental health disorders, making it a priority in the majority of government policies in Western countries, particularly in the wake of the COVID-19 pandemic [26]. Additionally, the notion of “necessary” may be attributed to the substantial body of scientific evidence supporting various mental health therapies, such as therapeutic exercise and body awareness therapies. This evidence includes numerous meta-analyses demonstrating the effectiveness of therapeutic exercise in various mental health conditions, including anxiety [27,28], depression [29], schizophrenia [30], eating disorders [31], and body awareness therapies [32].

Regarding the results of the tools used in PTMH, therapeutic exercise emerges as the most prominent, which aligns with the abundance of scientific evidence on a global scale. In a bibliometric study by Lidia Carballo [33], more than 19,588 publications on exercise in mental health were found. The most prevalent topics include eating disorders, addictions, chronic diseases, cancer, neurological problems, chronic pain, and severe mental disorders.

The next category that appears is the therapeutic relationship as a tool, and it is indeed a fundamental condition for interventions in PTMH, as it is not a mere intervention on its own. The therapeutic relationship is always present and gains even greater importance due to the relational characteristics of patients with mental disorders. One common feature of mental disorders is the distortion of reality and relationships with the environment. Therefore, the physiotherapist must be skilful in creating an environment where therapeutic relational factors can be manifested to generate adherence and acceptance of the proposed intervention. A study analysing adherence to physiotherapy treatment also highlights the importance of the therapeutic relationship as a crucial factor to consider [34].

As future lines of research stemming from this study, we could propose differentiating the perceptions of physiotherapists from those of the rest of the mental health professionals’ team. This could assist us in disseminating the role of physiotherapists in mental health, thereby enabling their inclusion in the team as a professional responsible for physical functionality and its implications on the symptoms of the patient’s mental disorders.

With all this information on the role of the physiotherapist in mental health in Spain, various actions are intended to be taken by the Spanish Association of Mental Health Physiotherapy. Given the limited awareness of the physiotherapist’s role in mental health, targeted awareness campaigns could be launched through the Association’s social media platforms. These campaigns would emphasize the concepts that we have identified as divergent from the role advocated by the European Region World Physiotherapy. Notably, these misconceptions involve exclusively addressing physical disorders or incorporating manual therapy into the toolkit. Another proposed action includes providing more specialized training to physiotherapists through high-quality courses, incorporating the latest evidence into the content.

## 5. Limitations

This study reveals inherent limitations of the qualitative methodology, where the interpretation of results is influenced by the authors’ prior knowledge of the subject. The interpretation of the qualitative results in this study was subject to the authors’ perspective on the specialization, such as their assessment of certain participants’ understanding regarding the objectives, functions, or tools used within mental health physiotherapy. However, a primary limitation was the professional profiles of participants. The main objective was to assess the knowledge of all transdisciplinary mental health professionals, but this goal was not realized, with physiotherapists constituting most of the professional profiles. This circumstance represents a gap, and further investigation is warranted to understand its implications for the study’s findings. It could be interesting to conduct this study within each professional domain in isolation to understand the perspectives of various mental health professionals on this role. Additionally, doubts are raised about the survey design and the respondents’ understanding, as many provided similar responses when asked about functions, objectives, and image. Nevertheless, the study provides a picture of the current state of the specialty, allowing for the development of policies to effectively establish it within our society.

## 6. Conclusions

In conclusion, the vision and role of this professional profile were unknown to the respondents. Despite being perceived as having a holistic view of the patient and being considered an essential need, the actual image remains vague. However, there is significant interest, indicating a promising future, although the lack of specialized training is noted. Therefore, the need for specialized education and awareness campaigns among professionals in the mental health field is highlighted. Additionally, promoting policies for the integration of this role within the multidisciplinary mental health team is essential.

## Figures and Tables

**Figure 1 healthcare-11-03136-f001:**
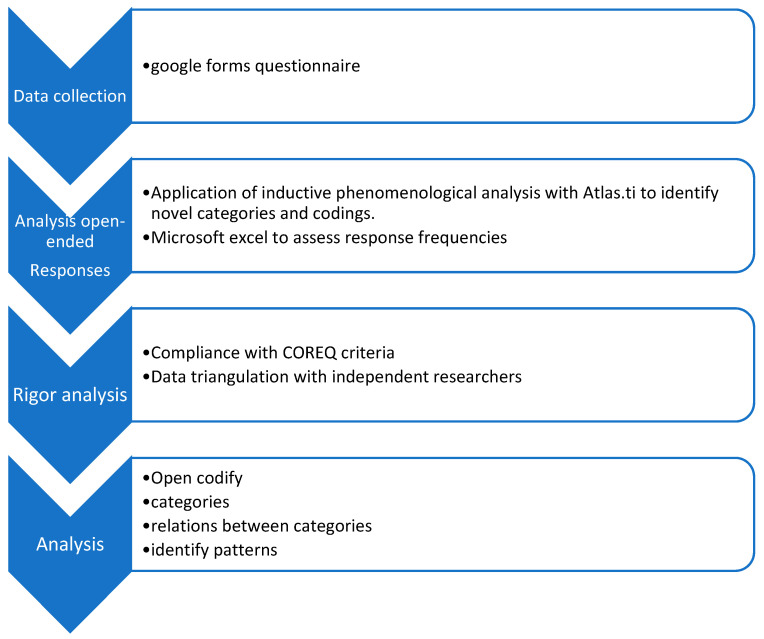
Methodology of analysis.

**Figure 2 healthcare-11-03136-f002:**
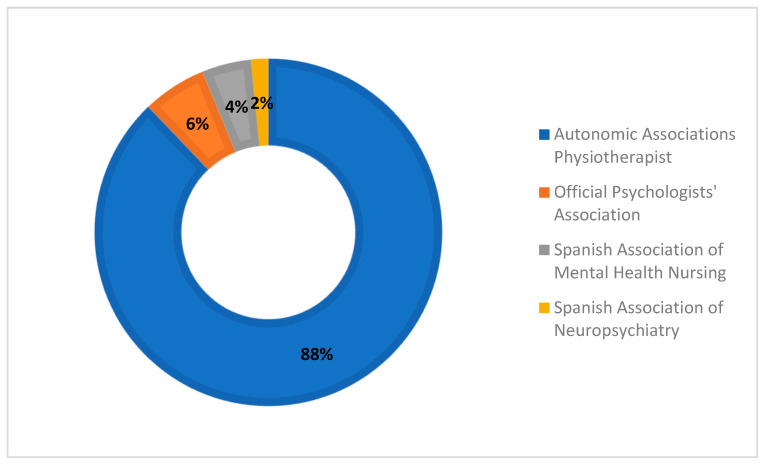
Flow chart of the professional associations in mental health that participated in the survey dissemination.

**Figure 3 healthcare-11-03136-f003:**
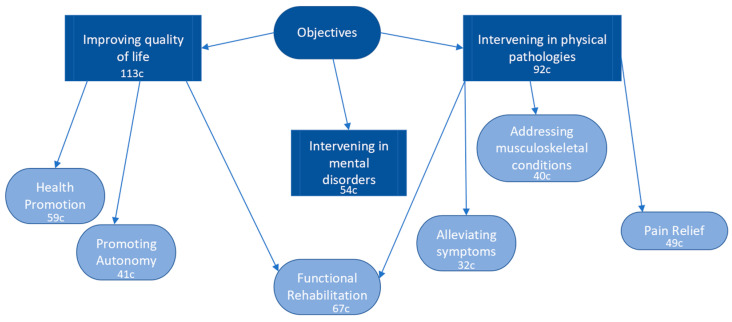
Results and patterns of the objectives of physiotherapy in mental health.

**Figure 4 healthcare-11-03136-f004:**
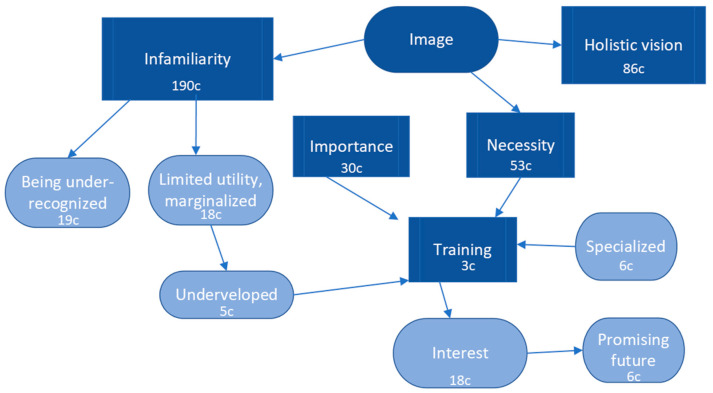
Results and patterns of the image of physical therapy in mental health.

**Figure 5 healthcare-11-03136-f005:**
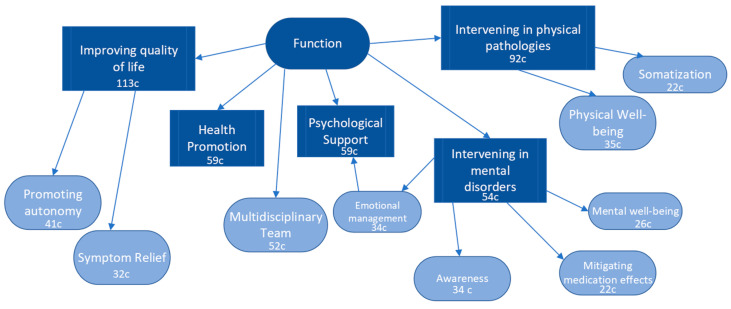
Results and patterns of the functions of physical therapy in mental health.

**Figure 6 healthcare-11-03136-f006:**
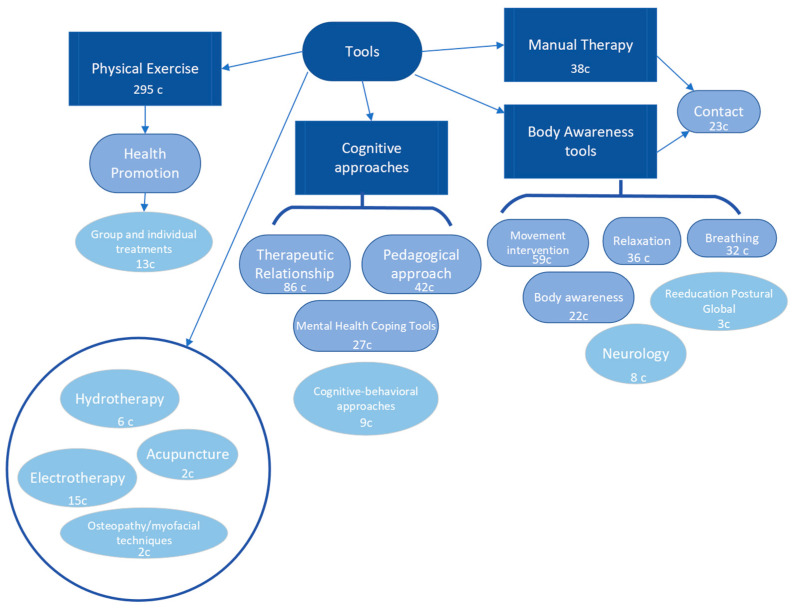
Results and pattern of the tools of physical therapy in mental health.

**Table 1 healthcare-11-03136-t001:** Structured questionnaire.

Variables	Questions
Sociodemographic Data	- Age - Gender - Profession
Professional Activity	- How many years of professional experience do you have?
	- Where do you currently work?
	- If you are a physiotherapist, what is your field of work? What type of patients do you treat in your practice?
Knowledge	- Are you aware of the existence of physiotherapy in the field of mental health?
	- Is there a role for a mental health physiotherapist in your multidisciplinary team?
	- If you answered yes, could you indicate the center where this role exists?
Education	- Do you have any training in mental health?
	- What kind of mental health training have you received?
	- Was the role of a mental health physiotherapist discussed during your undergraduate studies?
	- What about during postgraduate training?
Image	- How would you describe the image or general idea you have of mental health physiotherapy?
Role	- What do you believe is the role or function of a physiotherapist within the multidisciplinary team in the field of mental health?
Perception	- As a professional in a multidisciplinary mental health team, what is your perception of the role of the physiotherapist and their profession (physiotherapy) within your team?
Function	- What do you think is the purpose or function of physiotherapy in the field of mental health?
Objectives	- What are the current objectives of the work carried out by physiotherapists in mental health services? If you are not familiar with mental health physiotherapy, what do you think these objectives might be?
Tools	- What tools do you believe mental health physiotherapists use in their practice?
Necessity	- Do you think having a physiotherapist in a multidisciplinary mental health team is necessary?

**Table 2 healthcare-11-03136-t002:** Sociodemographic data and professional activity of participants.

Variable	Results
Age	37.5 [9.73]
Years of experience	14.33 [9.55]
Gender	78.4% women–21.6% men
Profession	88.3% Physical therapist
	6% Psychologist
	4.6% Nurse
Workplace	33.8% Private clinical centre
	14.7% Public hospital centre
	10.4% Socio-sanitary centre
	9% Primary care centre
	5.7% Public socio-sanitary centre
	3.5% Mental health centre
Work area	74.1% musculoskeletal physiotherapy
	43.5% neurology
	37.3% geriatrics
	22.2% paediatrics
	20.7% respiratory
	11.4% mental health
	9% gynecology

Note: Socio-sanitary centres include day care centres and nursing homes for the elderly and people with dependence. Mental health centres provide specific psychiatric and psychological care for people with mental health disorders, including prevention and rehabilitation programmes. They can be public or private.

**Table 3 healthcare-11-03136-t003:** Knowledge and training of the physiotherapist in mental health.

Questions	Results	
Are you aware of the existence of the MHPT?	56.8% yes	41.8% no
Does the figure of MHPT exist?	92% no	8% yes
Do you consider the figure of MHPT necessary?	78.9% “yes, very much”13% “is important but not necessary”	

Training in MH	69.5% no training30.5% yes, they have training	
-42% grade-8% postgraduate-6.9% master-2.3% PhD
Type of training	

Was the figure of MHPT addressed during the degree?	65.1% no17.6% yes17.3% no degree	


And in postgraduate training?	66.5% no	21.9% yes

Note: MHPT: physiotherapist in mental health.

**Table 4 healthcare-11-03136-t004:** Representative quotations for each code.

Objectives
“Helping improve your quality of life by bringing together the psychological and physical aspects.” (participant n.134)
“A therapy that helps patients enhance their quality of life through exercise, keeping both body and mind in check.” (participant n.89)
“Boosting mental health quality through body awareness and movement” (participant n.32)
**Image**
“Right now, I’m feeling a general sense of not being well-informed, and on a personal level, I’m really eager to dive deeper into the topic.” (participant n.302)
“I see it as a highly significant role within the multidisciplinary team, but unfortunately, it’s often underestimated by fellow physical therapists who don’t understand what we do, as well as by some other team members.” (participant n.167)
“It’s an underappreciated role with a lot of potential in treating mental disorders based on the scientific evidence in the field of physical therapy.” (participant n.198)
**Functions**
“Helping people with mental health issues integrate and adapt to their daily activities and participate in society.” (participant n.45)
“Enhancing the patient’s quality of life and assisting them in pursuing goals and motivations.” (participant n.59)
“Improving the patients’ quality of life by promoting greater physical autonomy and addressing the physical challenges associated with their specific disorders and treatment side effects.” (participant n.295)
“To enhance patients’ quality of life in terms of mobility, body awareness, and movement. To work alongside other professionals, such as psychologists, to provide a better understanding of their body, both externally and internally.” (participant n.242)
**Tools**
“Using therapeutic exercise and other tools to help patients maintain their quality of life. This will improve physical condition, stimulate cognitive function, and minimize the side effects of certain medications.” (participant n.88)
“Becoming more aware of the body and different sensory perceptions, participating in group exercise to enhance body image and social connections, and reaping benefits for the brain.” (participant n.351)
“The physical therapist who tackles mental illnesses using physiotherapy tools such as therapeutic exercise and physical agents.” (participant n.109)

## Data Availability

The article does not include material previously published. The authors give the permission to reproduce the article in the journal.

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
