# Peer review of "Beliefs and Self-Perceptions of Spanish Mental Health Professionals about Physical Therapy in Mental Health: An Observational Survey Study"

_healthcare, 2023, doi:10.3390/healthcare11243136_

Round 1

Reviewer 1 Report

Comments and Suggestions for Authors

Thank you for the opportunity to review your very interesting topic of research. It is highly important to have mental health working closely with physiotherapy given the statistics on life span and comorbidities.

Introduction:

I would like to have read more data from countries that are doing this well. Can you provide examples of where mental health and physiotherapy are working well together please?

Methodology/data analysis:

I would have liked to see some quotations from respondents. The themes told a great story, though actual quotes may provide more insight into the mindset of practitioners and thus how to meld mental health with physiotherapy successfully. Are you able to provide any that might elevate the voices of respondents.

Discussion:

One grammar incident:

P. 11. Line 321 - “Regarding the results regarding…” needs a different word than using regarding twice on one sentence.

Thank you for the privilege of reviewing your paper.

Comments on the Quality of English Language

One grammar incident:

P. 11. Line 321 - “Regarding the results regarding…” needs a different word than using regarding twice on one sentence.

Author Response

Comments Reviewer 1: Thank you for the opportunity to review your very interesting topic of research. It is highly important to have mental health working closely with physiotherapy given the statistics on life span and comorbidities.

Comment 1: Introduction: I would like to have read more data from countries that are doing this well. Can you provide examples of where mental health and physiotherapy are working well together please?

Answer 1: Thank you very much for your feedback. Based on your suggestion, we incorporated two sentences that explain some overview of the perceptions of physiotherapists in other countries. Please, find it below or in the manuscript in red color pag 3 line 101

Currently, specialties in physiotherapy exist in Austria, Denmark, Finland, Germany, Italy, the Netherlands, and United Kingdom. In this country, local areas of the National Health System (NHS) offers PTMH in their service directory(NHS Foundation Trust 2023). Both NHS and the Chartered Society of Physiotherapy (CSP) have published documents describing the requirements to become a specialist, and policy documents and treatment guides regarding PTMH(Health Education England 2023; The Chartered society of Physiotherapy 2023).

 and In pag 2 line 78:

According to studies conducted in Australia(Andrew et al. 2019; Ribeiro et al. 2022), physiotherapists feel prepared to manage the physical health of patients with mental disorders, although they request more undergraduate education on mental health and that the health system barriers are addressed. Another study(Stubbs et al. 2014) conducted with International Organization Physical Therapists in Mental Health (IOPTMH) members on the role of physiotherapy in the treatment of patients suffering from schizophrenia concludes that physiotherapists are an integral part of the multidisciplinary team that have a focused on promoting the physical health needs of these patients.

Comment 2: Methodology/data analysis: I would have liked to see some quotations from respondents. The themes told a great story, though actual quotes may provide more insight into the mindset of practitioners and thus how to meld mental health with physiotherapy successfully. Are you able to provide any that might elevate the voices of respondents.

Answer 2: We greatly appreciate your comments. Regarding this matter, it should be included a table with the quotations of the proposed themes. Please find it below or in the manuscript in red color in pag 7 line 219

Some of the most prominent responses are displayed in Table 4.

Table 4. Representative quotations for each code

Objectives

“Helping improve your quality of life by bringing together the psychological and physical aspects." (participant n.134 )

“A therapy that helps patients enhance their quality of life through exercise, keeping both body and mind in check." (participant n. 89)

“Boosting mental health quality through body awareness and movement” (participant n. 32 )

Image

“Right now, I'm feeling a general sense of not being well-informed, and on a personal level, I'm really eager to dive deeper into the topic." (participant n.302 )

"I see it as a highly significant role within the multidisciplinary team, but unfortunately, it's often underestimated by fellow physical therapists who don't understand what we do, as well as by some other team members." (participant n.167 )

"It's an underappreciated role with a lot of potential in treating mental disorders based on the scientific evidence in the field of physical therapy." (participant n.198 )

Functions

"Helping people with mental health issues integrate and adapt to their daily activities and participate in society.” (participant n.45 )

“Enhancing the patient's quality of life and assisting them in pursuing goals and motivations." (participant n.59 )

"Improving the patients' quality of life by promoting greater physical autonomy and addressing the physical challenges associated with their specific disorders and treatment side effects." (participant n.295 )

"To enhance patients' quality of life in terms of mobility, body awareness, and movement. To work alongside other professionals, such as psychologists, to provide a better understanding of their body, both externally and internally." (participant n. 242)

Tools

"Using therapeutic exercise and other tools to help patients maintain their quality of life. This will improve physical condition, stimulate cognitive function, and minimize the side effects of certain medications." (participant n.88 )

"Becoming more aware of the body and different sensory perceptions, participating in group exercise to enhance body image and social connections, and reaping benefits for the brain." (participant n. 351)

"The physical therapist who tackles mental illnesses using physiotherapy tools such as therapeutic exercise and physical agents." (participant n.109 )

Comment 3: Discussion: One grammar incident:

P .11. Line 321 - “Regarding the results regarding…” needs a different word than using regarding twice on one sentence.

Answer 3: Thank you very much for your attention and for notifying us of this minor incident. We were able to resolve it by replacing the second 'regarding' with the word 'for'. Please, find it below or in the manuscript in red color in pag 12 line 349.

Regarding the results of the tools used in physiotherapy in mental health…”

Thank you for the privilege of reviewing your paper.

Reviewer 2 Report

Comments and Suggestions for Authors

It was my pleasure to review the manuscript “Beliefs and Perceptions of Spanish Mental Health Professionals About Physical Therapy in Mental Health: An Observational Survey StudyThe aim of the study was to understand the beliefs held by professionals in the mental health field about the concept of mental health physiotherapy.

The abstract is well-written and concise.

 The introduction section is well written. Please check grammar ln 106-108, ln 133-135.

 Material and methods

What was the qualification of the researchers who conducted the analysis? (ln 171-172)

The role of the funding source (ln 185-186) should be moved to the end of the manuscript.

Figure 3 is confusing and requires explanation.

No statistical data were presented in the table summary, outside of demographic data and knowledge and training of the physiotherapists.  

The results are well-presented and instructive.

The discussion does not fully explain how utilizing the data extracted from the survey will help authors to contextualize the role of the physiotherapist.

Grammer ln 321-322

Limitations of the study addressed.

The statement “The study reveals inherent limitations of the qualitative methodology, where the interpretation of results is influenced by the authors' prior knowledge of the subject” was not established by previous data. 

Comments on the Quality of English Language

It was my pleasure to review the manuscript “Beliefs and Perceptions of Spanish Mental Health Professionals About Physical Therapy in Mental Health: An Observational Survey Study” The aim of the study was to understand the beliefs held by professionals in the mental health field about the concept of mental health physiotherapy.

The abstract is well-written and concise.

 The introduction section is well written. Please check grammar ln 106-108, ln 133-135.

 Material and methods

What was the qualification of the researchers who conducted the analysis? (ln 171-172)

The role of the funding source (ln 185-186) should be moved to the end of the manuscript.

Figure 3 is confusing and requires explanation.

No statistical data were presented in the table summary, outside of demographic data and knowledge and training of the physiotherapists.  

The results are well-presented and instructive.

The discussion does not fully explain how utilizing the data extracted from the survey will help authors to contextualize the role of the physiotherapist.

Grammer ln 321-322

Limitations of the study addressed.

The statement “The study reveals inherent limitations of the qualitative methodology, where the interpretation of results is influenced by the authors' prior knowledge of the subject” was not established by previous data. 

Author Response

Comments Reviewer 2:

It was my pleasure to review the manuscript “Beliefs and Perceptions of Spanish Mental Health Professionals About Physical Therapy in Mental Health: An Observational Survey Study” The aim of the study was to understand the beliefs held by professionals in the mental health field about the concept of mental health physiotherapy.

Comment 1: The abstract is well-written and concise.

Answer 1: Thank you very much, we are very glad that you liked it.

Comment 2:  The introduction section is well written. Please check grammar ln 106-108, ln 133-135.

Answer 2: We greatly appreciate your comments. Regarding this matter, it should modify some sentence in order to improve the grammar. Please find it below and in the pag 3 line 107 and line 134.

“PTMH has been evolving in recent years and includes a wide variety of evidence-based techniques(Catalán Matamoros 2019) such as physical activity or body awareness therapies, that have shown to improve both physical and mental symptomatology, as well as the quality of life for individuals with various types of mental disorders such as depression, anxiety, eating disorders or squizophrenia (Heywood et al. 2022; Catalan-Matamoros 2007; Vancampfort, Probst, and Skjaerven 2011).”

“This understanding will help to identify preconceived notions about this area and establish the foundations for its development in terms of employment, education, and research.”

Comment 3:  Material and methods: What was the qualification of the researchers who conducted the analysis? (ln 171-172)

Answer 3: Thank you very much for your feedback. Based on your suggestion, we incorporated a sentence that explain the qualification of the researchers. Please, find it below and in the pag 5 line 178.

These three researchers, who independently analyzed the data, are part of the research team for this study, and they are also doctoral candidates and physiotherapists with prior experience in qualitative studies and proficiency in using Atlas.ti.

Comment 4: The role of the funding source (ln 185-186) should be moved to the end of the manuscript.

Answer 4: We greatly appreciate your feedback and believe that, since the project has no funding, we can remove this section. At the end of the article, we explicitly mention that it is unfunded.

Comment 5: Figure 3 is confusing and requires explanation.

Answer 5: Thank you very much for your feedback. Based on your suggestion, we incorporated a sentence that explain the structure of the figure 3. Please, find it below and in the pag 9  line 256 .

“As can be seen in Figure 3, we establish relationships between the categories "unfamiliarity" and "limited utility, and marginated" since the latter factor is related to the limited development and the need for training required by this new specialization. On the other hand, many participants express their interest and see this specialization as having significant potential in the future”.

Comment 6: No statistical data were presented in the table summary, outside of demographic data and knowledge and training of the physiotherapists.  

Answer 6: We greatly appreciate your feedback and we would to explain that The quantitative data has been analyzed by presenting it as relative frequencies in percentage (%) based on the responses provided by the participants. No further statistical analyses such as means or standard deviations have been conducted. This approach allows you to present the distribution of responses or categories in proportional terms, which facilitates the understanding of how data is distributed across different categories or responses. We have incorporated the exact term into the text “relative frequencies in percentage (%)” in pag 5 line 170.

Comment 7: The results are well-presented and instructive.

Answer 7: Thank you very much for your feedback. We greatly appreciate.

Comment 8: The discussion does not fully explain how utilizing the data extracted from the survey will help authors to contextualize the role of the physiotherapist.

Answer 8: Thank you for your comment. We are confident that adding a sentence outlining future lines of action derived from the study will enhance the understanding of this research. Please find it below and pag 13 line 370.

“With all this information on the role of the physiotherapist in mental health in Spain, various actions are intended to be taken by the Spanish Association of Mental Health Physiotherapy. These actions include providing more specialized training to physiotherapists and, on the other hand, promoting awareness campaigns within the mental health professionals' community to disseminate this role”.

Comment 9: Grammer ln 321-322

Answer 9: Thank you very much for your attention and for notifying us of this minor incident. We were able to resolve it by replacing the second 'regarding' with the word 'for'. Please, find it below or in the manuscript in red color in pag 12 line 349.

Regarding the results of the tools used in physiotherapy in mental health…”

Comment 10: Limitations of the study addressed.

Answer 10: Thank you for your appreciation, your comment greatly encourages us.

Comment 11: The statement “The study reveals inherent limitations of the qualitative methodology, where the interpretation of results is influenced by the authors' prior knowledge of the subject” was not established by previous data. 

Answer 11: Thank you very much for your feedback. Based on your suggestion, we incorporated a sentence that explain the interpretation of results and their assessment. Please, find it below and in the pag 13 line 377.

“The interpretation of the qualitative results in this study is subject to the authors' perspective on the specialization, such as their assessment of certain participants' understanding regarding the objectives, functions, or tools used within mental health physiotherapy”.

Reviewer 3 Report

Comments and Suggestions for Authors

The purpose of this study is to understand the images, perceptions, and beliefs of the physical therapist's role in mental health physical therapy from professionals and other multidisciplinary teams. In the healthcare field, patient-centered, interdisciplinary collaboration is of great importance and requires mutual understanding among multiple professions. This study can be an essential resource for improving multidisciplinary cooperation in the medical field, which requires interdisciplinary collaboration.

Major Comments

In this study, the selection criteria, exclusion criteria, and study period regarding the research subjects are clearly stated. It is essential to clarify the status of the research subjects as it provides the rationale for the study. The criteria for setting the sample size in this study need to be clearly defined. It is also a critical item to indicate this information in the research paper in terms of the research process. Please clearly indicate the criteria for setting the number of research subjects (standards for sample size) in the methodology.

In addition to the above content, I recommend that the numerical changes of the research subjects during the study period be shown using a flowchart diagram. Although it is well presented in the text of the paper, giving the flow of the study in a diagrammatic form as a flowchart will help the reader to understand the analysis better. Please consider presenting the survey flow as a flowchart from the start of the study to determine the number of subjects to be analyzed.

"2.4. Data synthesis and Analysis" describes the process of data analysis. In this description, it is stated that three independent researchers did the initial study and that the next stage of analysis was performed after the fact by the research team. (If my interpretation is incorrect, please correct me.)

I recognize that the methodology of qualitative analysis is critical in this study. Therefore, you should indicate more clearly the number of people in charge of the investigation, the situation, etc., as to whether the analysis was conducted by three independent analysts again as a research team or by an independent research team etc. Please consider this.

Minor Comments

In Figure 2, Figure 3, Figure 4, and Figure 5, there needs to be more consistency in the format, color, etc. of the relationship diagrams. The reader's understanding would be improved if there were more uniformity. However, if the author intentionally changed the format of each figure, please forgive me.

Thank you in advance for your consideration of the above.

Author Response

Comments Reviewer 3:

The purpose of this study is to understand the images, perceptions, and beliefs of the physical therapist's role in mental health physical therapy from professionals and other multidisciplinary teams. In the healthcare field, patient-centered, interdisciplinary collaboration is of great importance and requires mutual understanding among multiple professions. This study can be an essential resource for improving multidisciplinary cooperation in the medical field, which requires interdisciplinary collaboration.

Major Comments

Comment 1: In this study, the selection criteria, exclusion criteria, and study period regarding the research subjects are clearly stated. It is essential to clarify the status of the research subjects as it provides the rationale for the study. The criteria for setting the sample size in this study need to be clearly defined. It is also a critical item to indicate this information in the research paper in terms of the research process. Please clearly indicate the criteria for setting the number of research subjects (standards for sample size) in the methodology.

 Answer 1: Thank you for your comment. The selection of participants was done conveniently as the survey was disseminated through various professional associations in psychology, psychiatry, nursing, etc., and through the different regional professional colleges in our country. Please find it below or in red color at pag 4 line 155

“The survey was sent to a convenience sample of professionals working in the field of mental health.”

Comment 2: In addition to the above content, I recommend that the numerical changes of the research subjects during the study period be shown using a flowchart diagram. Although it is well presented in the text of the paper, giving the flow of the study in a diagrammatic form as a flowchart will help the reader to understand the analysis better. Please consider presenting the survey flow as a flowchart from the start of the study to determine the number of subjects to be analyzed.

Answer 2: Thank you for your suggestion. We are confident that adding a flowchart about the participants in relation to the different associations where it was disseminated will significantly improve our article. Please, find it below or in pag 6 line196 and 209.

Figure 2. Flow chart of the professionals’ associations in mental health that participated in the survey dissemination.

Comment 3: "2.4. Data synthesis and Analysis" describes the process of data analysis. In this description, it is stated that three independent researchers did the initial study and that the next stage of analysis was performed after the fact by the research team. (If my interpretation is incorrect, please correct me.)

Answer 3: Thank you very much for your feedback, and we are confident that this comment will improve the understanding of the text. Indeed, we will modify the sentence to make it consistent with the paragraph. Please, find it below or in pag 5 line 182.

“The data was analyzed until saturation was reached by the three researchers, providing an interdisciplinary perspective, and reaching consensus on the results.”

Comment 4: I recognize that the methodology of qualitative analysis is critical in this study. Therefore, you should indicate more clearly the number of people in charge of the investigation, the situation, etc., as to whether the analysis was conducted by three independent analysts again as a research team or by an independent research team etc. Please consider this.

 Answer 4: Thank you very much for your feedback. Based on your suggestion, we incorporated a sentence that explain the qualification of the researchers. Please find it below or in red color at pag 5 line 178

“These three researchers, who independently analyzed the data, are part of the research team for this study, and they are also doctoral candidates and physiotherapists with prior experience in qualitative studies and proficiency in using Atlas.ti.”

Minor Comments

 Comment 5: In Figure 2, Figure 3, Figure 4, and Figure 5, there needs to be more consistency in the format, color, etc. of the relationship diagrams. The reader's understanding would be improved if there were more uniformity. However, if the author intentionally changed the format of each figure, please forgive me.

 Answer 5: Thank you very much for your feedback; we have taken your comment into account, and we have standardized the format of the figures and added the number of citations for each category.

Thank you in advance for your consideration of the above.

Reviewer 4 Report

Comments and Suggestions for Authors

Article "Beliefs and perceptions of Spanish mental health professionals about physical therapy in mental health: An Observational Survey Study", reports the results of an online survey about the opinion of physical therapy in mental health.

It is an interesting point of view, very much driven by an interest in supporting a physical intervention in mental health. However, the survey seems to be circular, as 88.3% of the participants are physiotherapists and most of them have no experience in mental health. So the respondents have an interest in giving importance to physical health, but lack expertise in this area, or at least it is not mentioned.

Perhaps it would be better to include self-perception in the title.

In summary, interesting but needs to be more precise and clearer.Perhaps it would be better to put self-perception in the title.

Impact statement,

Mention "this professional", be explicit about which profession

The perception of the role of the PT in mental health is essential, Add “for themselves”, it should be pointed out.

Points out that there are promising future prospects but there is no data to support this in this article.

Introduction

It should focus more on the Spanish reality, where the problem takes place. Perhaps make a comparison with an environment where it is diametrically different and where there is training, regulation and jobs for PTs in the context of mental health.

Synthesise the historical aspects, which are not the main part of the intro, but what is the reality of PT in mental health in Spain and what is it like in other countries where it is crucial.

Be more precise in the interventions and affections that could be addressed by physiotherapy, and what techniques and concrete actions, rather than everything general, in the end it is necessary to open the space with specific contents.

If you talk about Spain, mention it, avoid mentioning our country as in line 89.

Methodology

It is not entirely clear. Some sections are missing, e.g. instruments (the content is missing); procedure...

Concerning the instrument. There is a question in Table 3: "Do you think the

MHPT is necessary?", but it is not clear from the answers given what the gradient is. Better way of reporting the instrument

Table 1 uses a central justification that should be avoided.

Regarding the participants, it is not clear how many are students, even more than the perception of professionals in general or TF.

Results

In Table 2: Please explain at the bottom of the table what you mean by social and health centres, are they public, are they private, are they day-care centres, are they nurseries?

In Table 3, please explain the acronyms at the bottom of the table, e.g. MHPT.

Table 4 and the results in relation to the functions of physiotherapy in mental health are redundant as they repeat similar content.

In the figures, add the number of citations to make it easier to understand and the weight of each, do not repeat the same information in text and figure.

Discussion,

Put more emphasis on the weaknesses, on the absence of psychiatry professionals or other professions related to mental health (6% of psychologists and 4.6% of nurses, not mental health nurses).Point out that this is a view from the TF itself rather than from professionals in general.There are results that are lost for example line 2018-19 “the objectives seem to focus more on the intervention of physical pathologies rather than purely addressing mental disorders”, line 229 ....role is not well-known…

As a future line to compare between physio and non-physio the role of physiotherapy in mental health, even some data could be provided with this survey.References, check formal aspects, contains errors in general, for example lack of doi among many others.

Put more emphasis on the weaknesses, on the absence of psychiatrists or other professions related to mental health (6% of psychologists and 4.6% of nurses, not mental health nurses).Point out that this is a view from the TF itself rather than from professionals in general.There are results that are lost for example line 229 ....role is not known,

As a future line to compare between physio and non-physio the role of physiotherapy in mental health, even some data could be provided with this survey.

References

Check formal aspects, contains errors in general, for example lack of doi among many others.

Author Response

Comments Reviewer 4:

Article "Beliefs and perceptions of Spanish mental health professionals about physical therapy in mental health: An Observational Survey Study", reports the results of an online survey about the opinion of physical therapy in mental health.

Comment 1: It is an interesting point of view, very much driven by an interest in supporting a physical intervention in mental health. However, the survey seems to be circular, as 88.3% of the participants are physiotherapists and most of them have no experience in mental health. So the respondents have an interest in giving importance to physical health, but lack expertise in this area, or at least it is not mentioned.

Perhaps it would be better to include self-perception in the title. In summary, interesting but needs to be more precise and clearer. Perhaps, it would be better to put self-perception in the title.

Answer 1: We greatly appreciate your feedback, and indeed, the majority of the participants were physiotherapists. Nevertheless, our aim was to try to capture all opinions, but it was not possible as it did not have much reach among other professional profiles. Therefore, we have taken into account your suggestion to change the title of the article.

Comment 2: Impact statement. Mention "this professional", be explicit about which profession. The perception of the role of the PT in mental health is essential, Add “for themselves”, it should be pointed out.

Points out that there are promising future prospects but there is no data to support this in this article.

Answer 2: Thank you very much for your feedback. Based on your suggestion, we have modified the statement to make them more precise. Please find it below or in red color in pag 1 line 44

  • The respondents are unfamiliar with the vision and role associated with physiotherapist in mental health.
  • The perception of the role of the physical therapist in mental health is considered essential for themselves owing to their holistic patient perspective.
  • There is no sufficient specialized training in physiotherapy in mental health, despite the substantial interest shown by professionals working in this field.

Comment 3: Introduction. It should focus more on the Spanish reality, where the problem takes place. Perhaps make a comparison with an environment where it is diametrically different and where there is training, regulation and jobs for PTs in the context of mental health.

Answer 3: Thank you very much for your feedback. Based on your suggestion, we incorporated two sentences that explain some overview of the perceptions of physiotherapists in other countries. Please, find it below or in the manuscript in red color pag 3 line 101

Currently, specialties in physiotherapy exist in Austria, Denmark, Finland, Germany, Italy, the Netherlands, and United Kingdom. In this country, local areas of the National Health System (NHS) offers PTMH in their service directory(NHS Foundation Trust 2023). Both NHS and the Chartered Society of Physiotherapy (CSP) have published documents describing the requirements to become a specialist, and policy documents and treatment guides regarding PTMH(Health Education England 2023; The Chartered society of Physiotherapy 2023).

Comment 4: Synthesise the historical aspects, which are not the main part of the intro, but what is the reality of PT in mental health in Spain and what is it like in other countries where it is crucial.

Answer 4: We greatly appreciate your feedback, and we have condensed the PTMH's history section for better comprehension and alignment with the overall text.  Please find it below or pag 2 line 70.

“Physical and mental health conditions coexist, and they should not be treated isolated(Heywood et al. 2022). Psychiatry and mental health perception was influenced by philosophy and psychoanalysts, arising concepts like somatization, related to poor physical outcomes. Psychocorporal and somatic approaches were developed by Norwegian physiotherapists like Trygve Braatoy and Bülow-Hansen, pioneers in Nordic body awareness approaches such as "Norwegian Psychomotor Physiotherapy" and "Basic Body Awareness Therapy”, which are currently used in mental health physiotherapy (Michel Probst 2018). “

Comment 5: Be more precise in the interventions and affections that could be addressed by physiotherapy, and what techniques and concrete actions, rather than everything general, in the end it is necessary to open the space with specific contents.

Answer 5: Thank you very much for your comment, and in that regard, we have included new information in the introduction. Please, find it below or in pag 3 line 118.

“Mental health physiotherapy has been evolving in recent years and includes a wide variety of evidence-based techniques(Catalán Matamoros 2019) such as physical activity or body awareness therapies, that have shown to improve both physical and mental symptomatology, as well as the quality of life for individuals with various types of mental disorders such as depression, anxiety, eating disorders or squizophrenia (Heywood et al. 2022; Catalan-matamoros 2007; Vancampfort, Probst, and Skjaerven 2011; Fernández Cervantes et al., n.d.).”

Comment 6: If you talk about Spain, mention it, avoid mentioning our country as in line 89.

Answer 6: Thank you very much for your comment, and in that regard, we have modified those mentions to Spain.

Methodology

Comment 7: It is not entirely clear. Some sections are missing, e.g. instruments (the content is missing); procedure...Concerning the instrument. There is a question in Table 3: "Do you think the MHPT is necessary?", but it is not clear from the answers given what the gradient is. Better way of reporting the instrument

Answer 7: Thank you for your comment. We are confident that adding a sentence outlining about survey development and change the title of subsection 2.2, will enhance the understanding of this research. Please find it below and pag 4 line 159. About the question about the importance of physiotherapy in mental health, the survey had four possible responses: a) yes, very important, b) important but not necessary, c) neither important nor necessary, d) other response (this was open-ended). The results showed that only options a and b were chosen. Table 3 only displays the results obtained from the survey and not all the options. Nevertheless, if you deem it important, we can include a result with 0% indicating 'neither important nor necessary'.

2.2. Instrument and Data Extraction

The survey was designed through the consensus of various mental health experts who analyzed the different questions over a period of one month to address the research question. All relevant variables were identified, and open-ended questions were crafted to gather the maximum possible data.

Comment 8: Table 1 uses a central justification that should be avoided.

Answer 8: Thank you very much for your feedback. Based on your suggestion, we modify the table 1.

Comment 9: Regarding the participants, it is not clear how many are students, even more than the perception of professionals in general or TF.

Answer 9: Thank you very much for your feedback regarding the participants. Indeed, the initial idea was to include students, and we did, in fact, send the survey to several universities. However, we encountered various bureaucratic issues in being able to administer it in that population, and as a result, they did not respond. We have removed these participants from the methodology in the text, as they ultimately did not participate.

Results

Comment 10: In Table 2: Please explain at the bottom of the table what you mean by social and health centres, are they public, are they private, are they day-care centres, are they nurseries?

Answer 10: Thank you very much for your feedback. Based on your suggestion, we have added an explanation at the bottom of the table 2. Please find it below or in red color in pag 7 line 212

Note: *Socio-sanitary centers include day care centers and nursing homes for elderly and people with dependence. **Mental health centers provide specific psychiatric and psychological care for people with mental health disorders, including prevention and rehabilitation programs. They can be public or private.

Comment 11: In Table 3, please explain the acronyms at the bottom of the table, e.g. MHPT.

Answer 12: Thank you very much for your feedback. Based on your suggestion, we have added an explanation at the bottom of the table 3. Please find it below or in red color in pag 7 line 217

“Note: MHPT: physiotherapist in mental health”

Comment 12: Table 4 and the results in relation to the functions of physiotherapy in mental health are redundant as they repeat similar content.

Answer 12: Thank you very much for your comment, and indeed, the results in relation to the objectives and functions were very similar because the participants confused the two concepts. We initially considered removing the results for functions, but we believed that this data had been collected in the survey and should be presented.

 Comment 13: In the figures, add the number of citations to make it easier to understand and the weight of each, do not repeat the same information in text and figure.

Answer 13: Thank you very much for your feedback. We greatly appreciate. We changed all the figures with citation in each box and deleted from text. We are confident that this change will significantly improve the article.

Discussion,

Comment 14: Put more emphasis on the weaknesses, on the absence of psychiatry professionals or other professions related to mental health (6% of psychologists and 4.6% of nurses, not mental health nurses).Point out that this is a view from the TF itself rather than from professionals in general.There are results that are lost for example line 2018-19 “the objectives seem to focus more on the intervention of physical pathologies rather than purely addressing mental disorders”, line 229 ....role is not well-known…

Answer 14: Thank you very much for your feedback. Based on your suggestion, we incorporated a sentence for discuss that idea. Please find it below or in pag 13 line 344.

“In relation to the results of image regarding knowledge gaps, they align with findings from other studies wherein medical professionals exhibit a lack of knowledge within our field. Even final-year medical students in Nigeria display a moderate level of unfamiliarity along with a positive attitude towards mental health. Hence, numerous participants have articulated the notion that the specialty primarily involved the management of physical pathologies rather than mental health concerns.”

Comment 15: As a future line to compare between physio and non-physio the role of physiotherapy in mental health, even some data could be provided with this survey.

Answer 15: Thank you very much for your feedback. Based on your suggestion, we incorporated a sentence for discuss that idea. Please find it below or in pag 13 line 379.

“As future lines of research stemming from this study, we could propose differentiating the perceptions of physiotherapists from those of the rest of the mental health professionals' team. This could assist us in disseminating the role of the physiotherapist in mental health, thereby enabling inclusion in the team as a professional responsible for physical functionality and its implications on the symptoms of the patient's mental disorders.”

References

Comment 16: References, check formal aspects, contains errors in general, for example lack of doi among many others.

Answer 16: We greatly appreciate your comments. Regarding this matter, it should modify the references section.

Reviewer 5 Report

Comments and Suggestions for Authors

This is an exciting topic, covering the knowledge and perceptions of health professionals on the role of physical therapy in mental health. Patients who are dealt with physical therapy require minimal to very time-consuming assessment and treatment, in parallel to a mental health part of the condition (if in conjunction with another primary pathology) or as a separate pathology. 

Can you please provide more details on who constructed the questionnaire administered to health professionals?

Also, not all health professionals who filled in the questionnaire worked in mental health (only 11,4%). If not, were they aware of referring patients to specialists for mental health support in parallel to physical therapy related to 'mental health therapy'?

Finally, why not conduct this research only on physical therapists rather than on many health professionals who do not have specific knowledge of physical therapy content and code of practice?

Comments on the Quality of English Language

Good

Author Response

Comments Reviewer 5:

This is an exciting topic, covering the knowledge and perceptions of health professionals on the role of physical therapy in mental health. Patients who are dealt with physical therapy require minimal to very time-consuming assessment and treatment, in parallel to a mental health part of the condition (if in conjunction with another primary pathology) or as a separate pathology. 

Comment 1: Can you please provide more details on who constructed the questionnaire administered to health professionals?

Answer 1: Thank you for your comment. We are confident that adding a sentence outlining about survey development will enhance the understanding of this research. Please find it below and pag 4 line 163.

The survey was designed through the consensus of various mental health experts who analyzed the different questions over a period of one month to address the research question. All relevant variables were identified, and open-ended questions were crafted to gather the maximum possible data.

Comment 2: Also, not all health professionals who filled in the questionnaire worked in mental health (only 11,4%). If not, were they aware of referring patients to specialists for mental health support in parallel to physical therapy related to 'mental health therapy'?

Answer 2: Your comment regarding the professional profile of the participants is very interesting. We believe that currently, healthcare professionals are highly aware of mental health disorders and frequently refer these patients primarily to psychologists and psychiatrists. However, the model we advocate for is based on a multidisciplinary teamwork approach in which each professional profile possesses specific competencies. They can complement their treatments in this way, resulting in a comprehensive, patient-centered, and biopsychosocial approach. We believe that indeed patients are referred to these professionals, and that they are well taken care of by them. Although we believe that due to the limited number of physiotherapists working with a genuine focus on physiotherapy in mental health, patients are not being referred for movement-based interventions to improve mental health symptoms.

Comment 3: Finally, why not conduct this research only on physical therapists rather than on many health professionals who do not have specific knowledge of physical therapy content and code of practice?

Answer 3: Thank you very much for your comment. The goal of including all professional profiles working with mental health disorders was to gain an understanding of the perceptions of other members of the multidisciplinary team. This approach allows us to work on awareness campaigns regarding our role and promote better acceptance among other professionals.

Comments on the Quality of English Language

Good

Round 2

Reviewer 2 Report

Comments and Suggestions for Authors

Dear authors,

Thanks for addressing the reviewer's comments. Please check grammar line 105 "schizophrenia". The paper is in much better shape.

Comments on the Quality of English Language

Dear authors,

Thanks for addressing the reviewer's comments. Please check grammar line 105 "schizophrenia". The paper is in much better shape.

Author Response

I am pleased to resubmit for your consideration the revised version of the manuscript entitled “Believes and perceptions of Spanish mental health professionals about physical therapy in mental health: an observational survey study”.

A minor spelling error was noted by the reviewer and has been corrected. We hope that this article will be a great success for your journal and will get a significant number of readings and citations.

Your sincerely,

Silvia Solé

Reviewer 5 Report

Comments and Suggestions for Authors

The paper is sufficiently improved and can proceed to publication.

Comments on the Quality of English Language

Good

Author Response

I am pleased to resubmit for your consideration the revised version of the manuscript entitled “Believes and perceptions of Spanish mental health professionals about physical therapy in mental health: an observational survey study”.

We greatly appreciate your comments. We are very grateful for the excellent suggestions and comments from the reviewers, which have helped us in further improving the quality and clarity of this manuscript.

Your sincerely,

Sílvia Solé
